# Blood Pressure Elevation of Tubular Specific (P)RR Transgenic Mice and Lethal Tubular Degeneration due to Possible Intracellular Interactions between (P)RR and Alternative Renin Products

**DOI:** 10.3390/ijms23010302

**Published:** 2021-12-28

**Authors:** Sae Saigo, Tabito Kino, Kotaro Uchida, Takuya Sugawara, Lin Chen, Michiko Sugiyama, Kengo Azushima, Hiromichi Wakui, Kouichi Tamura, Tomoaki Ishigami

**Affiliations:** 1Graduate School of Medicine, Yokohama City University, Yokohama 236-0004, Japan; saesaigo1019@googlemail.com (S.S.); tabito.kino@temple.edu (T.K.); swinging_jazz_life@yahoo.co.jp (K.U.); takuya09061911617@hotmail.co.jp (T.S.); mysterylin@foxmail.com (L.C.); vn_nv2525smile@yahoo.co.jp (M.S.); azushima@yokohama-cu.ac.jp (K.A.); hiro1234@yokohama-cu.ac.jp (H.W.); tamukou@yokohama-cu.ac.jp (K.T.); 2Cardiovascular Research Center, Lewis Katz School of Medicine, Temple University, Philadelphia, PA 19140, USA

**Keywords:** prorenin receptor, urinary tubules, alternative renin, V-ATPase, hypertension

## Abstract

The prorenin/renin receptor ((P)RR) is a multifunctional protein that is widely distributed in various organs. Despite intensive research for more than 20 years, this receptor has not been fully characterized. In this study, we generated mice overexpressing the tubular epithelial (P)RR gene ((P)RR-TG mice) to test the previously reported functional role of (P)RR by Ramkumar et al. in 2015 using tubular specific (P)RR KO mice. (P)RR-TG mice were maintained and analyzed in individual metabolic cages and were administered angiotensin II blocker (ARB), direct renin inhibitor (DRI), and bafilomycin, that is, vacuolar ATPase (V-ATPase) antagonist. (P)RR-TG mice were hypertensive and had alkalized urine with lower osmolality and Na^+^ excretion. ARB and DRI, but not bafilomycin, concurrently decreased blood pressure. Bafilomycin acidized urine of (P)RR-TG mice, or equivalently this phenomenon restored the effect of overexpressed transgene, suggesting that (P)RR functioned as a V-ATPase in renal tubules. Afterall, (P)RR-TG mice were mated with alternative renin transgenic mice (ARen2-TG), which we identified as intracellular renin previously, to generate double transgenic mice (DT-TG). Lethal renal tubular damage was observed in DT-TG mice, suggesting that intracellular renin may be a ligand for (P)RR in tubules. In summary, (P)RR did not substantially affect the tissue renin-angiotensin system (RAS) in our model of tubular specific (P)RR gene over-expression, but alternative intracellular renin may be involved in (P)RR signaling in addition to conventional V-ATPase function. Further investigations are warranted.

## 1. Introduction

The renin-angiotensin system (RAS) plays pivotal roles in the maintenance and regulation of electrolyte and water homeostasis in higher vertebrates. Impairments of the RAS result in various cardiovascular abnormalities in humans. In addition to the classical RAS, which comprises angiotensinogen, renin, angiotensin-converting enzyme, and angiotensin II, Ludwig et al. initially discovered ATPase H^+^ transporting accessory protein 2 (ATP6ap2), which is an accessory protein of membrane-bound proton pump vacuolar ATPase, in 1998 [1]. In 2002, Nguyen et al. reported that the fragment was a component of a larger protein, the prorenin/renin receptor ((P)RR), which is a single-transmembrane receptor composed of 350 amino acids with a molecular mass of 30 kDa. The gene encoding the (P)RR is located on the short arm of the X chromosome [2]. (P)RR was first identified in human kidneys and initially attracted attention in the context of tissue RAS regulation. In 2004, Ichihara and colleagues reported that (P)RR was antagonized by decoy peptide mimicking the pro-segment of prorenin, thereby promoting generation of angiotensin I from angiotensinogen as a substrate for enzymatic reactions [3]. Binding of the (P)RR to renin or prorenin activates intracellular tyrosine phosphorylation-dependent pathways in a RAS-independent manner [4,5,6]. However, this peptide did not target prorenin at all, as recent structural analysis has revealed [7,8].

Subsequent studies revealed that the (P)RR is a multifunctional protein that is ubiquitously expressed in various organs, including the kidneys, heart, and brain [2]. At least three isoforms of (P)RR exist: the full-length protein, an NH3 -terminal, and a COOH-terminal fragment. (P)RR cleaves processing enzymes such as furin [9], ADAM19 [10], and site 1 protease (S1P) [11]. The NH3-terminal segment is secreted extracellularly as soluble (P)RR [9], while the COOH-terminal fragment, also known as M8.9 (ATP6ap2), functions as an accessory component of the vacuolar ATPase (V-ATPase) involved in lysosomal acidification and autophagy. Heart-specific knockout (KO) mice demonstrate embryonic lethality and vacuolar degeneration in cardiomyocytes [12], in contrast to expectations that these mice would exhibit fewer myocardial adaptations such as cardiac hypertrophy because they were initially considered to constitute a novel system of the RAS. These findings suggest that the (P)RR works predominantly as an accessory protein of V-ATPase in cardiomyocytes.

In earlier experiments, human (P)RR overexpression in rats showed hypertension, higher heart rate [13], and enhanced COX2 expressions with proteinuria in kidney [14], therefore enhanced (P)RR expressions might be involved in human cardiovascular diseases such as hypertension, and heart failure. However, Rosendahl et al. reported that constitutively overexpressed murine (P)RR in mice did not cause hypertension or cardiac and renal fibrosis [15], which was contradictive to previous researches. Although it remained inconclusive about the cause of difference of these consequences, one may argue that the overexpression of a protein foreign to the species investigated (human Atp6ap2 in rats) may easily result in artefacts than the overexpression of an endogenous protein [16]. To investigate the role of tubular (P)RR in blood pressure regulation, Ramkumar et al. generated an inducible renal tubule (P)RR KO mouse to avoid the lethal effects of (P)RR deletion on organ development [17]. (P)RR KO mice exhibited a similar blood pressure (BP) pattern to that of control mice under varying sodium intake; however, after Ang II infusion, their hypertensive response was attenuated due to reduced renal expression of renal ENaC. In control mice, but not in renal tubule (P)RR KO mice, prorenin stimulated ENaC activity in isolated cortical collecting ducts (CDs). Trepiccione et al. reported that there were no differences in BP, sodium excretion, or ENaC expression following Ang II administration between nephron-wide (P)RR KO mice and control animals [18].

However, the link between (P)RR and local RAS remains unclear despite extensive research over the past 20 years. In this study, we newly generated transgenic mice overexpressing (P)RR in tubular epithelial cells using the Ksp-Cadherin promoter, which will be biologically reciprocal mirror symmetry to the mice of Ramkumar et al., to investigate the roles of (P)RR in BP and tubular electrolyte homeostasis.

## 2. Results

We successfully generated (P)RR-TG mice and obtained three strains of the Ksp (P)RR transgene expressed in somatic cells. The relative expression of the (P)RR transgene was analyzed using quantitative RT-PCR, and mice with the highest transgene expression were used for subsequent experiments.

### 2.1. Histopathological Examination and Quantitative RT-PCR of (P)RR Transgene

No significant fibrotic changes, tubular alignment deformities, or histopathological abnormalities were observed in the kidneys of (P)RR-TG mice in hematoxylin and eosin (HE) stain and Masson’s trichrome stain (Figure 1). The (P)RR transgene was predominantly expressed in the medulla as indicated by (P)RR/Atp6ap2 immunostaining using anti-FLAG monoclonal antibodies (Figure 2). Quantitative RT-PCR results revealed greater (P)RR transgene localization in the medulla than in the cortex of the kidneys (Figure 3). These results indicated that the (P)RR transgene with the Ksp-Cadherin promoter was significantly more activated in the medullary tubules than in the cortical tubules. Accordingly, changes in urine parameters may have been due to (P)RR transgene overexpression in both medullary and cortical tubules.

### 2.2. Phenotypic Analyses of (P)RR-TG Mice

Experimental overview was shown in Figure 4. No significant differences were observed in body weight (BW) and kidney weight between (P)RR-TG mice and WT (wild type) mice (Table 1). Systolic blood pressure (SBP) was measured using the tail-cuff method on days 0, 5, and 10 with a non-invasive BP monitor (MK-2000, Muromachi Kikai Co., Ltd., Osaka, Japan). On day 10, SBP was lower in WT mice than in (P)RR-TG mice (94.4 ± 1.4 mmHg vs. 115.2 ± 1.1 mmHg, *p* < 0.001). ARB and DRI administration significantly reduced SBP in (P)RR-TG mice (86.3 ± 1.1 mmHg vs. 115.2 ± 1.1 mmHg and 84.5 ± 1.4 mmHg vs. 115.2 ± 1.1 mmHg, respectively; *p* < 0.001). Administration of bafilomycin (a V-ATPase inhibitor) did not alter SBP in WT mice (94.4 ± 1.4 mmHg vs. 97.6 ± 2.2 mmHg) or (P)RR-TG mice (115.2 ± 1.1 mmHg vs. 116.9 ± 2.0 mmHg, *p* = n.s.) (Figure 5). We measured plasma renin activity (PRA) for both WT and (P)RR-TG mice. We could not find significant differences between two groups about PRA (*p* = 0.096). (PRA in WT (*n* = 3) is 9.80 ng/mL/h, and in (P)RR-TG (*n* = 5) is 20.70 ng/mL/h, respectively).

Water intake (WI) was measured daily. WI was significantly lower in (P)RR-TG mice than in WT mice (0.26 ± 0.01 g/BW day vs. 0.34 ± 0.01 g/BW day, *p* < 0.001) (Figure 6). There were no significant differences in WI between ARB-, DRI-, and bafilomycin-administered (P)RR-TG mice and untreated mice (0.25 ± 0.01 g/BW day vs. 0.26 ± 0.01 g/BW day, 0.30 ± 0.01 g/BW day vs. 0.26 ± 0.01 g/BW day, and 0.28 ± 0.01 g/BW day vs. 0.26 ± 0.01 g/BW day, respectively; *p* = n.s.).

### 2.3. Urine Volume, pH, Osmolality, and Na Excretion

UV were increased in (P)RR-TG mice compared to those of WT mice. (39.5 ± 2.5 g/BW day vs. 51.5 ± 2.5 g/BW day, *p* < 0.05.) (Figure 7). ARB, DRI, and bafilomycin administration did not significantly affect UV in (P)RR-TG mice (63.2 ± 3.6 g/BW day vs. 51.5 ± 2.5 g/BW day, 58.4 ± 4.5 g/BW day vs. 51.5 ± 2.5 g/BW day, and 42.1 ± 3.2 g/BW day vs. 51.5 ± 2.5 g/BW day, respectively; *p* = n.s.). Bafilomycin administration significantly increased UV in WT mice (54.3 ± 3.2 g/BW day vs. 39.5 ± 2.5 g/BW day, *p* < 0.01).

Urine pH was significantly higher in (P)RR-TG mice than in WT mice (7.8 ± 0.2 vs. 6.8 ± 0.2, *p* < 0.001) (Figure 8). ARB and DRI administration did not significantly affect urine pH in (P)RR-TG mice (7.8 ± 0.2 vs. 7.8 ± 0.2 and 7.7 ± 0.2 vs 7.8 ± 0.2, respectively; *p* = n.s.). Although bafilomycin administration restored these effects and acidized urine in (P)RR-TG mice (7.0 ± 0.1 vs. 7.8 ± 0.2, *p* < 0.05), this intervention was not effective in WT mice (7.5 ± 0.2 vs. 6.8 ± 0.2, *p* = n.s.).

Urine osmolality was lower in (P)RR-TG mice than in WT mice (2787.8 ± 161.5 mOsm/kg vs. 4043.3 ± 398.1 mOsm/kg, *p* < 0.05) (Figure 9). ARB, DRI, and bafilomycin administration did not significantly affect urine osmolality in (P)RR-TG mice (2613.8 ± 205.7 mOsm/kg vs. 2787.8 ± 161.5 mOsm/kg, 2719.2 ± 268.9 mOsm/kg vs. 2787.8 ± 161.5 mOsm/kg, and 2839.4 ± 268.9 mOsm/kg vs. 2787.8 ± 161.5 mOsm/kg, respectively; *p* = n.s.). Bafilomycin administration did not significantly affect urine osmolality in WT mice (2922.3 ± 334.7 mOsm/kg vs. 4043.3 ± 398.1 mOsm/kg, *p* = n.s.).

Urine Na^+^ excretion was significantly lower in (P)RR-TG mice than in WT mice (145.6 ± 8.4 mEq/L day vs. 199.4 ± 25.0 mEq/L day, *p* < 0.05) (Figure 10). ARB, DRI, and bafilomycin administration did not significantly affect urine Na+ excretion in (P)RR-TG mice (98.4 ± 10.7 mEq/L day vs. 145.6 ± 8.4 mEq/L day, 118.8 ± 13.6 mEq/L day vs. 145.6 ± 8.4 mEq/L day, and 149.0 ± 14.5 mEq/L day vs. 145.6 ± 8.4 mEq/L day, respectively; *p* = n.s.). Bafilomycin administration significantly decreased urine Na^+^ excretion in WT mice (131.5 ± 6.7 mEq/L day vs. 199.4 ± 25.0 mEq/L day, *p* < 0.05).

### 2.4. Histological Findings in Kidneys of Double Transgenic Mice

(P)RR-TG mice and ARen2-TG mice [19], which we identified as intracellular renin previously, were mated to generate double transgenic mice. Even after successful birth, a large proportion of offspring died shortly after birth (156 out of 353, i.e., 44.2% from 1 April 2019 to 31 December 2019). We investigated the cause of mortality by histopathological examination of fetuses. Fetuses were fixed with formalin and sliced into sections. Sections were stained with HE using standard methods. We observed extensive tubular degeneration in the cortico-medullary junction in the kidneys (Figure 11), suggesting that ARen2 and (P)RR interacted as ligand-receptors in the kidneys, resulting in lethal kidney disfunction.

## 3. Discussion

In our current experiments, we investigated newly developed (P)RR tubular transgenic mice in detail to verify Ramkumar’s hypothesis [17]. We generated transgenic mice overexpressing (P)RR in renal tubules using the Ksp-Cadherin promoter, which was a generous gift from Prof. Igarashi at Texas University. Using metabolic cage experiments with (P)RR-TG mice, we demonstrated that BP of (P)RR-TG mice was significantly higher than that of WT mice, and administration of ARB and DRI successfully reduced SBP in both (P)RR-TG mice and WT mice. ARB/DRI is an antihypertensive drug and has a blood pressure lowering effect independent of any pathophysiological background of hypertension. Therefore, even if both drugs are effective, it cannot be said that RAS is activated to develop hypertension. Both ARB and DRI are RAS inhibitors but target distinct aspects of the RAS. ARB inhibits the AT1 receptor, resulting in a positive feedback loop that increases renin/prorenin activity. In contrast, DRI binds to the active center of renin and inhibits renin activity, thereby causing a decrease in renin/prorenin activity. Therefore, if (P)RR undertakes the role in RAS these two agents may have acted differently in (P)RR-TG mice, but our results suggest that (P)RR does not function as a receptor of renin/prorenin, consistent with conventional theories. Recently, it has been reported that megalin in proximal tubules is working as a novel endocytic receptor for renin and prorenin. In vitro experiments discovered that endocytosis of renin and pro-renin by megalin is accompanied with (P)RR by Sun et al. [20] and others [21,22], suggesting limited roles of (P)RR for renin/prorenin binding in tubules. This evidence indirectly might be relevant to our findings that (P)RR and renin/prorenin are tenously related physiologically.

Blood pressure measurements might be influenced by stress (metabolic cages) and osmotic minipumps, which produce a pro-inflammatory state that may interfere with some parameters. Blood pressure data without minipumps and without metabolic cages would be nice, but currently we could not provide this to show these procedures’ impact on measurements of blood pressure. Thus, our experiments about blood pressure measurements have such a limitation.

Based on an in vitro study by Advani et al. [23], we administered (P)RR-TG mice with bafilomycin, a V-ATPase inhibitor. Administration of bafilomycin did not alter BP in (P)RR-TG mice. This result, therefore, cannot be explained by V-ATPase overexpression alone; variations in ENaC expression may be associated with this phenomenon in accordance with the report of Ramkumar et al. [17]. In addition to BP changes, our examination of urine pH revealed that urine was alkalized in (P)RR-TG mice, which could have been due to overexpression of V-ATPase. This was supported by the observation that administration of bafilomycin restored pH in the urine of (P)RR-TG mice. In addition, urine osmotic pressure was lower in (P)RR-TG mice and was not affected by bafilomycin administration. Urine Na^+^ concentration was lower in (P)RR-TG mice, and administration of bafilomycin decreased urine Na^+^ concentration in WT mice; however, no significant changes were observed in the urine of (P)RR-TG mice administered with bafilomycin. These results suggest that the (P)RR transgene functions primarily as a V-ATPase rather than part of RAS. Although we attempted to derive direct evidence of elevated BP in (P)RR-TG mice using quantitative RT-PCR of ENaC, we did not observe any significant difference in expression between (P)RR-TG mice and WT mice (data not shown). Detailed experimental procedures such as qRT-PCR experiments using laser-capture micro-dissection procedure are warranted to address this question.

We previously reported on the use of intracellular renin, otherwise known as alternative renin (ARen), transgenic mice [19]. To examine possible intracellular interactions between alternative renin and (P)RR, we performed mating experiments with ARen2-TG mice and (P)RR-TG mice and generated double transgenic (DT) mice. DT mouse fetuses developed extensive damage in the renal cortico-medullary region, in which both ARen2 and (P)RR transgenes are selectively expressed along the tubular region. Based on our histopathological observations (Figure 11), we hypothesized that ARen2 (intracellular renin) may function as a ligand for (P)RR within tubular epithelial cells. There are not enough detailed protein interaction experiments between ARen2 protein and (P)RR protein so far. Our experiment used genetical engineering techniques to artificially force gene expression of both (P)RR and ARen2 in tubular epithelial cells. This is a study that artificially examined the function of each protein using these animals. Although the precise molecular mechanisms remain unclear, we aim to clarify the direct relationship between intracellular renin and (P)RR in future studies.

Finally, in our original experiments in (P)RR-TG mice, we found that overexpression of (P)RR in renal tubules resulted in hypertension in mice in a RAS-independent manner, which is in accordance with the findings of Ramkumar et al. [17]. From urine characteristics, the function of V-ATPase was augmentated along with overexpressed (P)RR transgene on renal tubules, and antagonist of V-ATPase counteracted the effect of overexpressed transgene. The transgene on tubules alkalized urine and decreased osmotic pressure and Na^+^ concentration, which originally allowed us to elucidate an uncovered and unknown function of the (P)RR gene in tubules. Meanwhile, results of blood pressure do not imply that (P)RR acts merely as a V-ATPase in tubules. However, the relationship between BP elevation and (P)RR remains unclear. Although we were unable to resolve the ongoing debate regarding (P)RR, our findings highlight the possibility that (P)RR and ARen2 bind each other within epithelial tubular cells, based on high mortality in fetuses and their characteristic histopathological findings in DT mice. Identification of real interactions such as in vitro protein combinations between these two players may reveal insight into the mechanisms involving tissue RAS. Alternative renin, which we discovered as intracellular renin, may be involved in (P)RR signaling other than conventional renin/prorenin, but this requires further investigations.

## 4. Materials and Methods

### 4.1. Generation of Tubular Epithelial (P)RR Gene Overexpression Ksp-(P)RR Transgenic Mice

To investigate the role of the (P)RR gene in vivo and the pathological implications of the tubular (P)RR gene, we generated transgenic mice harboring the (P)RR gene with the Ksp-Cadherin promoter [24] by genetically engineering mice with (P)RR cDNA in the pKsp/BGH vector, which was a generous gift from Prof. Igarashi, Texas University. Constructs of transgene (P)RR with the Ksp-Cadherin promoter are shown in Figure 12.

(P)RR transgenic mice were generated at TransGenic Inc^®^ (Kumamoto, Japan) using our custom-made pENTR TOPO-SD-D/(P)RR vector provided to the company. The DNA fragment containing the (P)RR protein-coding region (1053 bp), and DYKDDDDK tagged at the 3′terminus (1080 bp) was amplified. These PCR products were inserted into the mal cloning site of the pKsp/BGH vector. To confirm the construction of the expression vector, the product was digested with restriction enzymes and electrophoretically fractionated, resulting in the construct. Subsequently, the transgene was injected into 200 prenuclear-stage C57BL/6 mouse embryos. Among these embryos, we selected functionally developed embryos and transplanted them into recipient ICR mice. After 20 days of pregnancy, mice with (P)RR gene overexpression along tubules were obtained. To ensure that these mice expressed the transgene (Ksp/BGH-(P)RR), PCR analysis was conducted after ablation using the caudae. The primers for PCR were primer F1, which amplified most of the transgene (3.1 kbp) and bound to the Ksp promoter (5′-CTCTAGAACTAGTGGATCCCCCAG-3′) and BGH-R1, which bound to BGH polyA (5′-TGGCAACTAGAAGGCACAGTCGAGG-3′).

### 4.2. Histopathological Examinations and Quantitative Analyses of Ksp-(P)RR-FLAG in Kidneys

For histological examinations, mouse tissues were fixed in 4% paraformaldehyde, embedded in paraffin, and sliced into 4-μm-thick sections. The sections were stained with HE and Masson’s trichrome stain using standard methods. In situ immunohistochemical analyses were performed using 1:500 dilution of anti-FLAG monoclonal antibodies (monoclonal ANTI-FLAG M2, Clone M2, SIGMA-ALDRICH^®^, Tokyo, Japan) to determine the distribution of transgene products, (P)RR-FLAG, and fusion protein. Morphological analysis was performed using a BZ-900 fluorescence microscope (Keyence, Osaka, Japan). Subsequently, the expression of the (P)RR transgene in mouse kidney was examined by qRT-PCR with GAPDH as an internal control using ΔΔCT methods with the following forward and reverse primers: 5′-TATGATCGGCTTGGCCTTGG-3′ and 5′-TCGAATCTTCTGATTTGTCAT CCT-3′. Quantitative RT-PCR experiments for transgene expressions were examined separately for the cortex and medulla of the mice kidneys. At the time of sacrifice, a fresh kidney was soaked in a petri dish and manually excised under a stereomicroscope, and, subsequently, total RNA was extracted from both the medulla and the cortex of part of the mice for experiments.

### 4.3. Metabolic Cage Analyses

Age-matched (10-week-old) male Ksp-(P)RR TG mice (*n* = 25) and WT mice (*n* = 7) were maintained for 10 days in individual metabolic cages (SN-781, Shinano Manufacturing Co., Ltd., Tokyo, Japan) under controlled conditions of light, temperature, and humidity after a 5-day habituation period (Figure 4). Animals were divided into six groups and received a normal salt diet followed by three types of medication: (1) WT (WT group, *n* = 3), (2) (P)RR TG ((P)RR group, *n* = 5), (3) (P)RR-TG with olmesartan treatment ((P)RR ARB group, *n* = 4, olmesartan 3 mg/kg/day), (4) (P)RR-TG with aliskiren treatment ((P)RR DRI group, *n* = 4, aliskiren 25 mg/kg/day), (5) WT (WT group, *n* = 4) administered with bafilomycin via an osmotic mini-pump (Alzet^®^, osmotic mini-pump) (WT Baf group, *n* = 4, bafilomycin 0.03 μg/g/day), and (6) (P)RR-TG administered with bafilomycin via an osmotic mini-pump (Alzet^®^, osmotic mini-pump) ((P)RR Baf group, *n* = 5, bafilomycin 0.03 μg/g/day).

Suitable drug regimens were determined according to a previous study. BW and WI were measured, and urine was collected daily. Urine sodium concentrations were measured using specific electrodes (Oriental Yeast Co., Ltd., Tokyo, Japan). Urine osmolality was determined using a cryoscopic osmometer (Oriental Yeast Co., Ltd., Tokyo, Japan). Urine pH was measured using an electrode pH meter (LAQUIAtwin^®^, pH-11, Horiba, Kyoto, Japan). BP was measured 10–15 times using the tail-cuff method on days 0, 5, and 10 with a noninvasive BP monitor (MK-2000, Muromachi Kikai Co. Ltd., Osaka, Japan). On day 11, the mice were sacrificed under isoflurane anesthesia, and blood samples were collected by cardiac puncture. Organs were dissected and fixed for subsequent total RNA extraction, protein extraction, immunoprecipitation, and histological analyses.

### 4.4. Generation of DT Mice

We previously generated transgenic mice with alternative renin (ARen2) overexpression in epithelial cells with the CAGGS promoter in 2014 [19]. To examine intracellular interactions between ARen2 and (P)RR, we performed mating experiments between (P)RR-TG and ARen2-TG to generate DT mice. We analyzed survival rates after birth and performed histopathological analyses of newborn DT mice using a BZ-900 fluorescence microscope (Keyence, Osaka, Japan).

### 4.5. Statistical Analyses

Data are expressed as the mean ± standard error (SE). All statistical analyses were performed using BellCurve for Excel (Social Survey Research Information Co., Ltd.) and EZR (Saitama Medical Center, Jichi Medical University, Saitama, Japan), which is a graphical user interface for R (The R Foundation for Statistical Computing, Vienna, Austria). The relative abundance of mouse transcripts was statistically compared using one-way ANOVA followed by Bonferroni’s multiple-comparisons test. Statistical significance was set at *p* < 0.05.

## 5. Conclusions

Although we did not fully elucidate the function of (P)RR, our results suggest that (P)RR may be irrelevant to RAS or it may function as a V-ATPase in renal tubules. DT-TG mice revealed ((P)RR-TG and ARen 2-TG mice) a possible interaction between (P)RR and intracellular alternative renin.

## Figures and Tables

**Figure 1 ijms-23-00302-f001:**
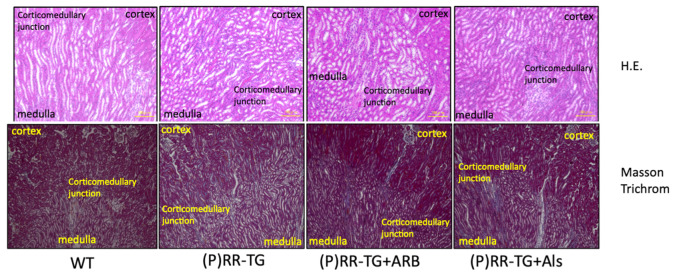
Histopathological findings using HE (**above**) and Masson trichrome (**below**) staining with X100 magnification by BZ-9000 (Keyence, Osaka, Japan).

**Figure 2 ijms-23-00302-f002:**
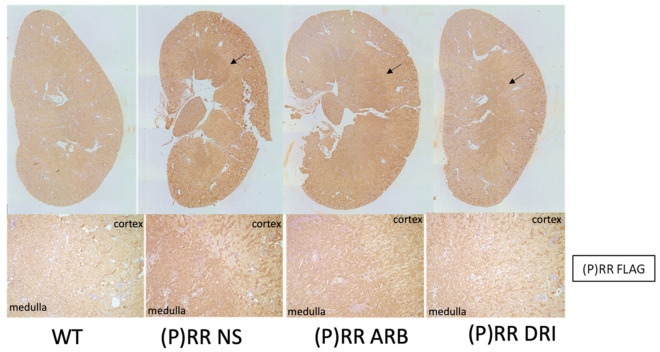
Representative histopathological findings using immunohistochemistry with anti-FLAG antibody (monoclonal ANTI-FLAG M2, Clone M2, SIGMA-ALDRICH^®^, Tokyo, Japan) are shown. Arrows indicate positive staining in lower magnification (**above**) and in higher magnification (X100, **below**). These findings were shown as WT, (P)RR-TG, (P)RR-TG+ARB, and (P)RR-TG+DRI, respectively.

**Figure 3 ijms-23-00302-f003:**
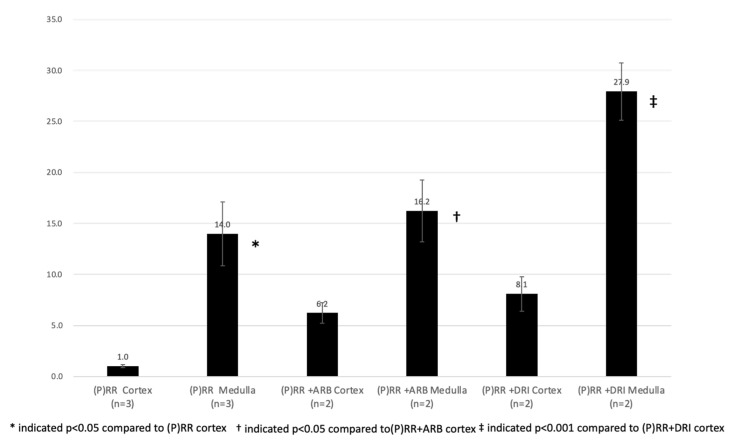
Expressions of (P)RR transgene relative to (P)RR cortex were shown. Results of qRT-PCR of (P)RR transgene in the kidneys of (P)RR-TG mice. Expression levels of transgenes were higher in the renal medulla than in the cortex. Data were analyzed using one-way ANOVA.

**Figure 4 ijms-23-00302-f004:**
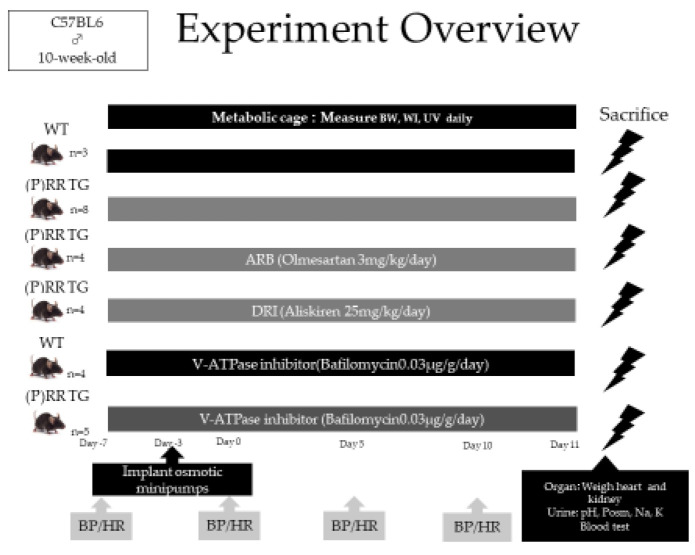
Experimental overview is shown. WT: wild type; (P)RR TG: (P)RR transgenic mice; BW: body weight; WI: water intake; UV: urine volume; ARB: angiotensin receptor blocker; DRI: direct renin blocker; BP: blood pressure; HR: heart rate.

**Figure 5 ijms-23-00302-f005:**
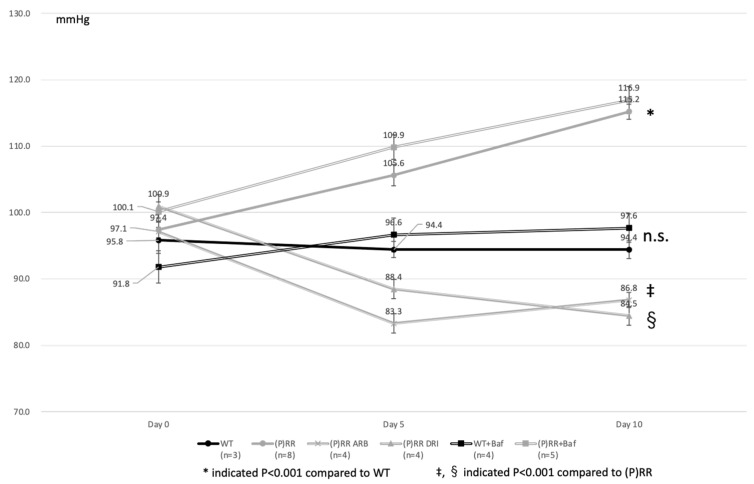
Blood pressure (mmHg) of animals is shown. Systolic blood pressure (SBP) was measured using the tail-cuff method on days 0, 5, and 10 with a non-invasive BP monitor (MK-2000, Muromachi Kikai Co., Ltd., Osaka, Japan). Data were analyzed using one-way ANOVA between (P)RR mice and *t*-test between WT and (P)RR mice. “n.s.” means “there is no significant difference between blood pressure of WT and that of WT+Baf.

**Figure 6 ijms-23-00302-f006:**
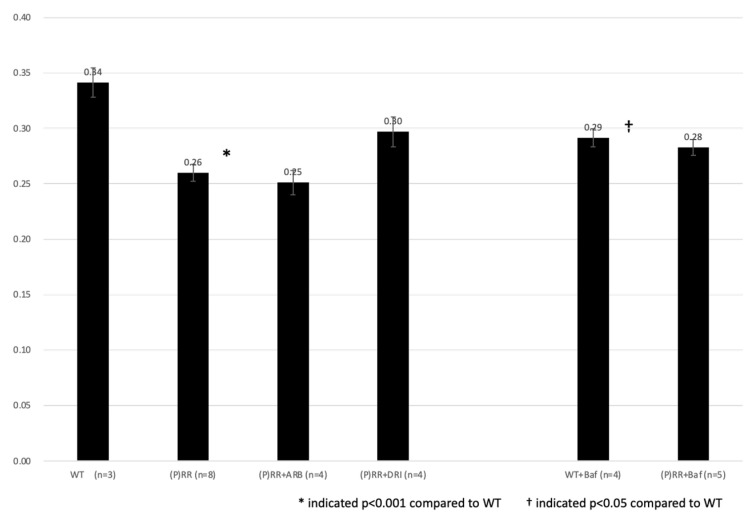
Average water intake (g/BW day) of animals were shown. Data were analyzed using one-way ANOVA between (P)RR mice and *t*-test between WT and (P)RR mice.

**Figure 7 ijms-23-00302-f007:**
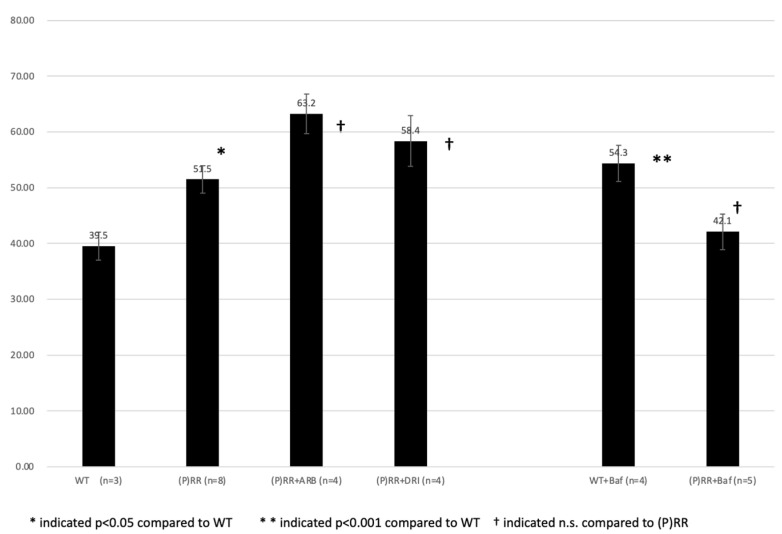
Average urine volume (UV) (g/BW day) of animals were shown. Data were analyzed using one-way ANOVA between (P)RR mice and *t*-test between WT and (P)RR mice.

**Figure 8 ijms-23-00302-f008:**
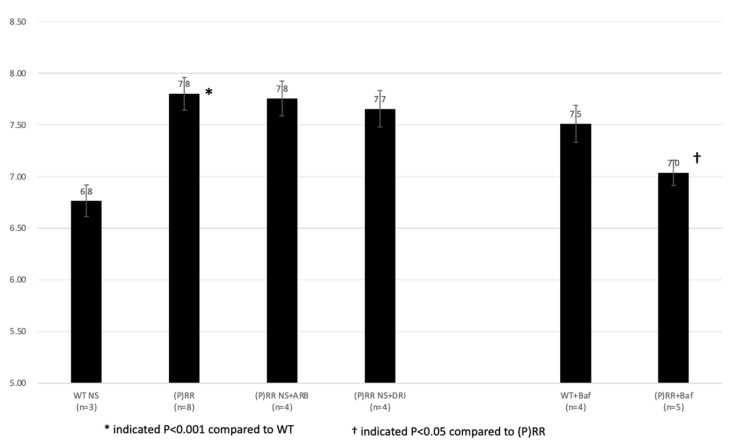
Average urine pH of animals is shown. Data were analyzed using one-way ANOVA between (P)RR mice and *t*-test between WT and (P)RR mice.

**Figure 9 ijms-23-00302-f009:**
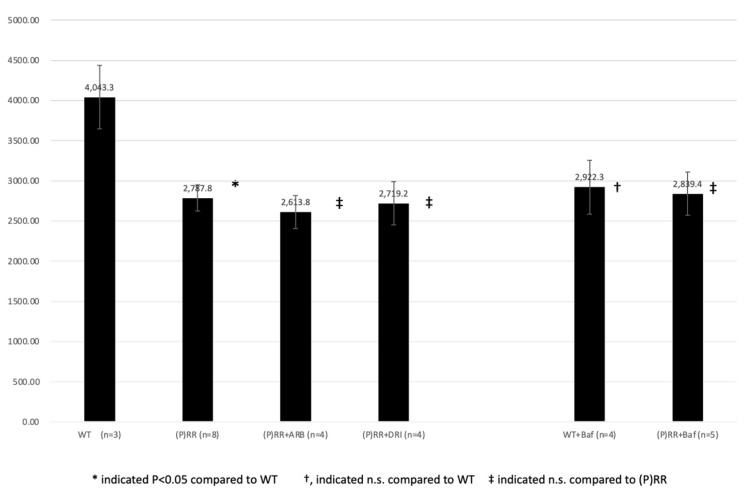
Urine osmolality (mOsm/kg) of animals is shown. Data were analyzed using one-way ANOVA between (P)RR mice and *t*-test between WT and (P)RR mice.

**Figure 10 ijms-23-00302-f010:**
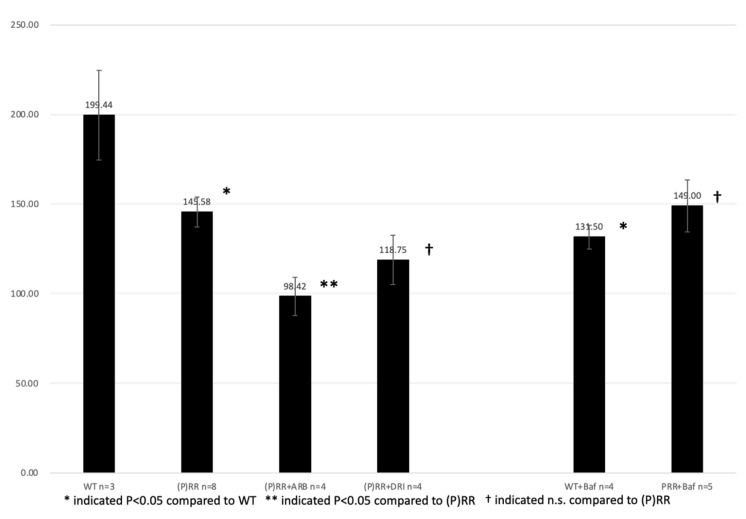
Average urine Na^+^ excretion of animals is shown. Data were analyzed using one-way ANOVA between (P)RR mice and *t*-test between WT and (P)RR mice.

**Figure 11 ijms-23-00302-f011:**
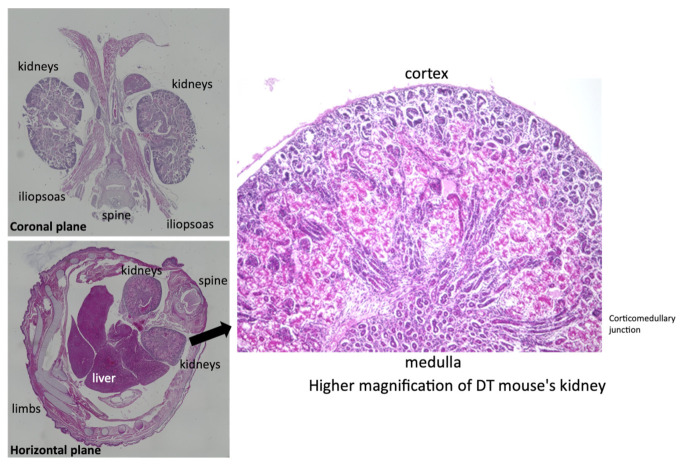
Representative histopathological ARen2-TG/(P)RR-TG DT (double transgenic) mice fetuses, by HE staining, at both lower magnifications of coronal plane, horizontal plane, and higher magnifications (X400).

**Figure 12 ijms-23-00302-f012:**
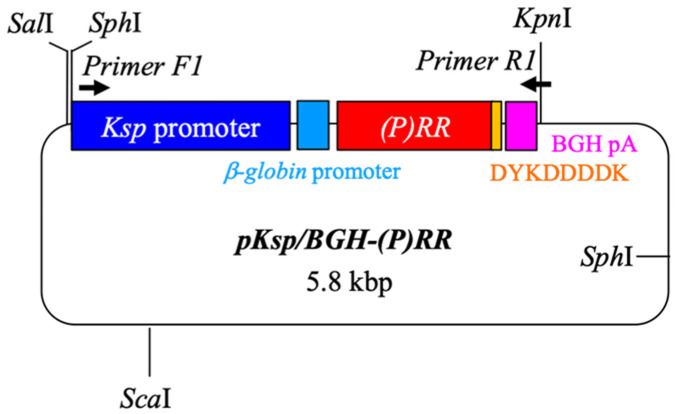
Construct of transgene.

**Table 1 ijms-23-00302-t001:** Summary of body weight and kidney weight of mice.

(g)		WT (*n* = 3)	(P)RR (*n* = 8)	(P)RR+ARB (*n* = 4)	(P)RR+DRI (*n* = 4)	WT+Baf. (*n* = 4)	(P)RR+Baf. (*n* = 5)	*p*
Body Weight	Day 0	20.6 ± 1.6	26.0 ± 1.5	26.0 ± 1.4	23.7 ± 2.4	22.2 ± 1.2	24.9 ± 0.5	N.S.
	Day 10	21.5 ± 1.8	27.2 ± 1.4	26.7 ± 1.6	25.9 ± 1.9	22.3 ± 1.1	25.7 ± 1.1	N.S
Kidney Weight		0.17 ± 0.02	0.18 ± 0.01	0.19 ± 0.01	0.18 ± 0.02	0.15 ± 0.00	0.15 ± 0.01	N.S

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
