# Peer review of "Blood Pressure Elevation of Tubular Specific (P)RR Transgenic Mice and Lethal Tubular Degeneration due to Possible Intracellular Interactions between (P)RR and Alternative Renin Products"

_ijms, 2021, doi:10.3390/ijms23010302_

Round 1

Reviewer 1 Report

  1. General comments:

The present manuscript describes the effects of renal tubular cell-specific overexpression of Atp6ap2, previously named (pro)renin receptor, on water intake, urine volume, pH, osmolarity, sodium excretion, and blood pressure. There was a marked increase in blood pressure that was independent of v-ATPase inhibition but could be reduced to reference values by inhibitors of the renin-angiotensin system. In addition, the authors observed severe tubular degeneration and increased lethality after crossbreeding these mice with transgenic mice overexpressing an alternative renin transcript (ARen2) on the same C57Bl6 background.

Most of the experiments deal exclusively with the effect of Atp6ap2 overexpression in renal tubular cells of mice and not with ARen2. These data alone are of great interest. With respect to the proposed interactions between Atp6ap2 and an alternative renin protein ARen2, however, the article does not hold what the title promises. Interactions between alternative renin products and the (pro)renin receptor have not been investigated at all. Just to crossbreed two strains of animals and describe some pathological changes does not allow any conclusion.

After having found the original manuscript by the authors about the identification of alternative renin transcripts in mice (Hypertension 2014; 64:125-133, which was not given in the reference list), I would like to summarize as follows with respect to any possible interaction of ARen2 and Atp6ap2:

1) No data have been provided so far demonstrating any interactions between the alternative renin isoform ARen2 and Atp6ap2 (neither in Hypertension 2014 nor here).

2) All renin proteins are certainly located intracellularly at some point (either within secretory vesicles or elsewhere). For ARen2 the location may be cytosolic. However, no data have been provided so far, demonstrating that ARen2 protein is located within the cytosol. Moreover, no data have been provided so far showing that ARen2 is at the same place than Atp6ap2 protein. In fact, most of the Atp6ap2 protein is known to be bound at membranes with the renin binding site pointing to the extracellular space or to the lumen of intracellular vesicles. Only a small fraction may be located completely within the cytosol (Kanda et al) and only this part may be acxcessible for ARen2.

3) No data have been provided so far demonstrating that ARen2 cannot be secreted.

4) No data have been provided showing that Atp6ap2 protein in the transgenic model cannot be cleaved to sPRR and secreted thus allowing extracellular effects and interactions with any form of renin.

4) No data have been provided demonstrating that Atp6ap2 functions are modulated by ARen2

  1. Specific comments to the experiments and analysis:

1) Statistics: ANOVA has not been applied correctly. There are three preconditions to compare groups using ANOVA:

  1. a) Normal distribution. This cannot be evaluated with n=3, however one may assume its presence.
  2. b) Equal variances. This may not be always given. Thus, in Table 1, day 0, the SE for body weight is between 0.5 and 2.4 = about fivefold difference of SE = probably great difference of variances.
  3. c) Groups that cannot be compared need to be excluded from ANOVA. Thus, one cannot compare untreated WT with PRR transgenic mice treated with ARB and so on. WT can only be compared with untreated PRR mice (i.e. Table 1) and the correct statistic for this comparison would be t-test (or Mann-Witney U test / rank test). Then, I assume, that there will be a significant difference between untreated WT and untreated PRR mice with respect to body weight (and so on...).

ANOVA would apply for all PRR-groups (excluding WT). Please recalculate for all figures and compare only those groups where only 1 parameter was different.

2) The fact that ARB or DRI decreases blood pressure in PRR mice does not allow the conclusion that the cause of hypertension is an activated RAS. RAS-Inhibitors will also decrease blood pressure if hypertension has developed RAS independently.

3) Atp6ap2 (PRR) protein data are missing (westerns, preferentially after subcellular fractionation), including full lenght PRR, sPRR and M8.9. Also with respect to Fig. 9, an anti-PRR antibody should be used as well to estimate the total amount of PRR protein (endogenous and transgene) and its location.

4) Blood pressure measurements are likely influenced by stress (metabolic cages) and osmotic minipumps produce a pro-inflamatory state that may interfere with some parameters. Thus, blood pressure data without minipumps and without metabolic cages would be nice – but at least this problem needs to be mentioned.

5) I have a question regarding the Ksp1 promoter: can activation of this promoter in the brain – and thus overexpression of Atp6ap2 in the brain be excluded? This would be essential since Atp6ap2 in the brain affects blood pressure.

6) Please include and discuss the paper by Rosendahl et al showing no renal dysfunction with ubiquitous overexpression of the PRR.

7) Please include and discuss the papers of Figuerida (Eur J Physiol 2021; 473:1229-1246), Sun (Hypertension 2029; 75:1242-1250) and Pohl ( JBC2010;285:41935-46) with respect to the role of Atp6ap2 in megalin mediated tubular uptake of proteins including components of the RAS.

8) Is there any information about the degree of autophagy in the model?

III. Further aspects:

Ref 33 does not provide information about the ARen2 transgenic rats. It should be Hypertension 2014; 64:125-133.

I would recommend to show the results for the evaluation of PRR transgenic expression first and then its consequences.

Table 1: it is not clear to which groups the p values refer

Figure 1: There are 7 “lightnings” but only 6 mice signs

Reviewer 2 Report

Title: Lethal intracellular interaction between alternative renin products and pro-renin receptor in tubular epithelium

Remarks to the author:

In this single study, the authors have generated mice overexpressing the tubular epithelial (P)RR gene using tubular specific (P)RR KO mice and were administered olmesartan (ARB), aliskiren (DRI), and bafilomycin, a vacuolar ATPase (V-ATPase) antagonist with the objective to investigate the functional role of (P)RR. The results showed that (P)RR-TG mice were hypertensive and had alkalized urine with lower osmolality and Na+ excretion compared to the WT mice whereas ARB and DRI, decreased blood pressure and bafilomycin acidized urine of (P)RR- TG mice. Also, the authors have generated double transgenic mice (DT-TG) by mating (P)RR-TG mice with alternative intracellular renin transgenic mice (ARen2-TG). They have observed lethal renal tubular damage around 44% of the population and thus they suggest that intracellular renin may be a ligand for (P)RR in tubules. In conclusion authors has mentioned that (P)RR did not substantially affect the tissue renin-angiotensin system (RAS), but alternative intracellular renin may be involved in (P)RR signalling in addition to conventional V-ATPase function, but further investigations are warranted.

Specific comments:

Introduction:

  1. (P)RR is a multifunctional protein express in various organs. However, the authors haven’t mentioned the specific reasons to investigate the importance of prorenin/renin receptor, especially in pathological conditions like hypertension or heart failure where RAS plays a major role. Therefore, the objective of this study needs to be justified.

Materials and methods:

  1. The authors have only measured the gene expression of (p)RR transgene. Since the blood is collected from mice at the end of the study, measuring renin in blood would be ideal to observe the outcome of (P)RR transgene overexpression in mice compared to the WT mates.
  2. Although the authors have mentioned that they have measured BP on day 0,4,7 and 10, the figure 2 shows the data for day 0, 5 and 10. Please revise accordingly.
  3. Immunohistochemical staining details including dilutions/concentrations needs to be explained or referred to the relevant literature.

 Results:

  1.  Figure 1: please expand the abbreviations at the end of the figure legend. Also, the figure needs to be revised and clearly point out (using arrows) the groups. The four boxes at the bottom of the figure with BP/HR needs to be explained and mentioned in the description. Please check the procedures mention in the box at the right bottom where additional procedures are mentioned other than the ones those have been explained in the manuscript.
  2. Figure 2: As per previous comment BP for mentioned time points needs to be added to the graph. Also, graph axes need to be labelled.
  3. Figure 3: Line 144-145 - According to the graph WI in WT mice is higher than that of (P)RR-TG mice. Please revise accordingly.
  4. Figure 8 and 9: Images needs be labelled to enable identification of regions observed. Especially in figure 9, positive immunohistochemical staining needs to be pointed to be identified as the staining is too faint.
  5. Figure 10: Number of animals used for PCR quantification is different from the numbers used for other analysis in the manuscript. The reasons for using less numbers need to be explained. Also, X axis labels are very hard to read. So please include a clear graph.
  6. Figure 11: Different regions in the kidney needs to be pointed out enabling to identify the lesions.

Overall, authors have repeated the description under each analysis in the figure legend. The result interpretation style needs to be changed.

Discussion

  1.  The introduction and the discussion got repetitive information (eg: Lines 255-265). Please revise accordingly

General comments:

  1. Figure labels are not clear.
  2. Histology images need to be labelled.

Round 2

Reviewer 1 Report

1) The authors answered most of my comments successfully. However, I still feel that the conclusion goes a little too far, although they are now somewhat “milder”. This refers also to the title.

The authors at least need to demonstrate that the level of transgenic overexpression of AREN-2 is similar in Ksp-1 driven AREN-2 transgenic mice and double transgenic mice. It is known from the literature that Ksp1 is regulated in response to autophagy-processes. Thus, one cannot rule out that AT6ap2 overexpression, which may have affected autophagy processes, increased KSP1-driven transgenic AREN-2 expression. If so, a stronger overexpression of AREN-2 may have produced the pathological effects already alone without interacting with ATP6Ap2. Therefore measurements and comparison of tubular AREN-2 mRNA levels in single AREN-2 transgenic and double transgenic mice would be desired to exclude this possibility.

2) The now observed significant differences between WT and TG (urine osmolarity, ph etc.) should be discussed (i. e. role of ATP6AP2 on urine concentration, pH etc.) more strongly. So far, only knockdown models are available showing the effect of ATP6ap2 absence on urine concentration or ph. The present manuscript thus provides important new/additional data and information to this field. Maybe the recent review by Hoffmann&Peters (Functions of the (pro)renin receptor (Atp6ap2) at molecular and system levels: pathological implications in hypertension, renal and brain development, inflammation, and fibrosis; Pharmacol Res 173:105922; doi: 10.1016/j.phrs.2021.105922.) could be included, however, this is just a suggestion, not a requirement.

Minor:

Lines 46-48: Here the observations by Rosendahl et al need to be mentioned as well.

Line 145: The star (*) in figure 7 indicated that there was a significant difference of urine volume between TG and TG (Column 2 vrs 1). However, in the text is written that there was no difference.
